



# Joint 1DVar Retrievals of Tropospheric Temperature and Water Vapor from GNSS-RO and Microwave Radiometer Observations

Kuo-Nung Wang[1], Chi O. Ao[1], Mary G. Morris[1], George A. Hajj[1], Marcin J. Kurowski[1], Francis J. Turk[1], and Angelyn W. Moore[1]

[1]Jet Propulsion Laboratory, California Institute of Technology, Pasadena, California, 91109, United States

**Correspondence:** Kuo-Nung Wang (Kuo-Nung.Wang@jpl.nasa.gov)

**Abstract.**

Global Navigation Satellite System – Radio Occultation (GNSS-RO) and Microwave Radiometry (MWR) are two of the most impactful spaceborne remote sensing techniques for numerical weather prediction (NWP). These two techniques provide complementary information about atmospheric temperature and water vapor structure. GNSS-RO provides high vertical reso-

lution measurements with cloud penetration capability, but the temperature and moisture are coupled in the GNSS-RO retrieval process and their separation requires the use of a-priori information or auxiliary observations. On the other hand, the MWR measures brightness temperature ($T_b$) in numerous frequency bands related to the temperature and water vapor structure, but is limited by poor vertical resolution (>2km) and precipitation.

In this study we combine these two technologies in an optimal estimation approach, 1D Variation method (1DVar), to better

characterize the complex thermodynamic structures in the lower troposphere. This study employs both simulated and operational observations. GNSS-RO bending angle and MWR $T_b$ observations are used as inputs to the joint retrieval, where bending can be modeled by an Abel integral and $T_b$ can be modeled by a Radiative Transfer Model (RTM) that takes into account atmospheric absorption, and surface reflection and emission. By incorporating the forward operators into the 1DVar method, the strength of both techniques can be combined to bridge individual weaknesses. Applying 1DVar to the data simulated from

Large Eddy Simulation (LES) is shown to reduce GNSS-RO temperature and water vapor retrieval biases at lower troposphere, while simultaneously capturing the fine-scale variability that MWR cannot resolve. A sensitivity analysis is also conducted to quantify the impact of the a-priori information and error covariance used in different retrieval scenarios. The applicability of 1DVar joint retrieval to the actual GNSS-RO and MWR observations is also demonstrated through combining collocated COSMIC-2 and Suomi-NPP measurements.

*Copyright statement.* © 2022. California Institute of Technology. Government sponsorship acknowledged





## 1 Introduction

Atmospheric profiles of temperature and water vapor are critical geophysical variables to various weather and climate processes. Humidity in the lower troposphere (LT), especially the planetary boundary layer (PBL), determines the strength and depth of convection and influences circulations and cloudiness, and indirectly affects atmospheric circulations through non-
local influence on infrared irradiances (Stevens et al , 2017). The thermodynamic effect of atmospheric moisture plays an important role in strong positive feedback on global warming and increased subtropical dryness (Sherwood et al, 2010), and negative feedback through low cloud formation (Mülmenstädt et al, 2021; Zhou et al, 2016). Also, surface air temperature and atmospheric water vapor content are found in Fujita and Sato (2017) to be connected to extreme precipitation under a warming climate. In particular, Holloway and Neelin (2009) shows that the vertical structure of the specific humidity, especially in the
free troposphere, is highly correlated to moist convections and rainfalls. To provide accurate measurements of temperature and water vapor, the Global Navigation Satellite System – Radio Occultation (GNSS-RO), passive Microwave Radiometer (MWR), and hyperspectral Infrared Sounders (IR) are the key spaceborne sounding techniques for numerical weather prediction and atmospheric science research. In this study we focus on the combination of GNSS-RO with MWR over IR because of simplicity (lower number of channels) and applicability (better penetration below the thick cloud).

GNSS-RO is a remote sensing technique used to observe the vertical thermodynamic structure from the bending of the occulted GNSS signal ray paths that propagate through the stratified atmosphere (Kursinski et al , 1997). By measuring the time-varying phase of the received signals, the bending angle of each ray path can be computed and inverted to retrieve the vertical profiles of refractivity, which can be used to retrieve temperature and moisture with a-priori information. With its limb sounding geometry, GNSS-RO is capable of providing global coverage with high vertical resolution (∼200m) observations. In
addition, the L-band navigation signals used in GNSS-RO can penetrate heavy cloud cover and precipitation and is independent of surface emissivity. Numerous GNSS-RO missions have been launched since 1995 and the number of RO observations has now reached more than 10,000 per day, including the ones from COSMIC-2 (July 2019), METOP-C (November 2018), GRACE Follow-on (May 2018), Sentinel-6 Michael Freilich (November 2020), and commercial CubeSats that have been deployed in recent years.

In addition to GNSS-RO, MWR is also commonly used to profile the atmosphere. MWR utilizes multiple frequency bands, providing information that can be related to the vertical structure of temperature and water vapor in the atmospheric column (Rodgers, 2000). The temperature profile is mostly linked to the oxygen absorption frequency bands located between $50-60$ GHz while the water vapor profile can be retrieved from the brightness temperature measurements of 23.8 GHz and 183 GHz water vapor absorption lines (Liu et al , 2021). Typically, the atmospheric temperature and water vapor profiles are retrieved
along with the surface temperature and emissivity simultaneously through an optimal estimation approach (Boukabara et al , 2011). This technology has been extensively utilized in both ground-based and airborne platforms, and numerous MWR instruments have been launched to low earth orbit (LEO) including the Advanced Microwave Sounding Unit (AMSU)/ Microwave Humidity Sensor(MHS) on NOAA-18, NOAA-19, Metop-A, Advanced Microwave Scanning Radiometer for Earth Observing System (EOS) (AMSR-E) on Aqua, and Advanced Technology Microwave Sounder (ATMS) on Suomi-NPP. Assimilation of





observation from AMSU/MHS (Bao et al , 2015), AMSR-E (Kazumori et al , 2008), ATMS (Bormann et al , 2013) in numerical weather prediction (NWP) models has shown significant positive impacts.

    While both techniques provide high quality observations, each has its own limitations. For example, MWR sounding suffers from poor vertical resolution (>2km) due to limited number of frequency channels and broad range of weighting functions. MWR measurements from different bands could also be affected by clouds, precipitation, aerosol, absorption by ozone and

carbon-dioxide, and surface properties, which introduce notable uncertainties in the MWR retrievals. Meanwhile, even though GNSS-RO refractivity and bending angle profiles are closely related to the vertical temperature and moisture structures, they cannot be used to retrieve temperature and water vapor independently without the use of auxiliary information, as explained in Sec. 2 below. In general, atmospheric profiles from a global weather analysis, such as NCEP or ECMWF, are utilized as a-priori states, which could be biased or erroneous and significantly impact the final temperature and moisture products derived

from the RO refractivity. In addition, the RO refractivity retrieval itself could be negatively biased due to ducting (Xie et al , 2006; Ao , 2007), which will be translated into considerable positive bias in the temperature and negative bias in the moisture at the top of boundary layer. While several bias-correction methods for RO within PBL have been proposed (Xie et al , 2006; Wang et al , 2020), they remain challenging to apply in practice.

    In this study, an 1-dimension variational (1DVar) estimation approach is implemented to combine GNSS-RO and MWR

measurements and simultaneously retrieve temperature and water vapor profiles that preserve the high vertical resolution from GNSS-RO. The idea of combining RO and MWR/IR soundings is not new: several methods have been proposed to take advantage of these complementary measurements. For example, von Engeln et al (2001) applied an optimal estimation method to determine the temperature profile from RO bending angle measurements and oxygen line radiance from the passive microwave limb sounder (MLS). In Borbas et al (2003) the multivariable regression method is used to estimate the atmospheric

states based on the coefficients trained by a set of RO refractivity and MWR/IR brightness temperature observations. The simulation results around the the tropopause altitude shows that the RO - IR combination can improve the temperature and moisture retrievals by 0.5K and 2.5%, respectively, compared to retrievals from IR alone. Ho et al (2007) took a similar regression approach and expanded the application to the lower troposphere to reduce the impacts of RO refractivity retrieval bias by introducing AIRS observations. For both studies, the the regression coefficients are calculated from specific training

datasets, which could be biased based on its spatial and temporal distribution. The non-linear behavior between the profile variables (temperature and water vapor) and the observables (brightness temperature and refractivity) could also induce errors in the simple linear regression expressions.

    The simulation study conducted in Collard and Healy (2003) shows that by combining RO and nadir sounding (MWR/IR) measurements using a sequential 1DVar algorithm, two complementary observations can be fused together and "contribute the

greatest impacts to different parts of atmospheric temperature and humidity fields." Collard and Healy (2003) performs 1DVar twice to include the background temperature and moisture information with RO refractivity and IR brightness temperature observations separately. While this approach works well for unbiased measurements, it is known that in the lower troposphere GNSS-RO refractivity observation could be negatively biased due to ducting which occurs at the top of PBL (Ao , 2007), phase unwrapping error caused by low SNR conditions (Wang et al , 2016), and possibly small-scaled refractivity fluctua-



tions(Gorbunov et al , 2015). To reduce the refractivity bias due to ducting, Wang et al (2017) developed an optimal estimation algorithm to choose the unbiased profile from a family of solutions with collocated AMSR-E total precipitable water (TPW) retrievals. However, the Wang et al (2017) study focuses on correcting the bias due to ducting with TPW retrieval, but does not utilize the full information provided by MWR observations.

In this study, we correct the RO bias and improve the retrieval for both RO and MWR with a more optimized approach by: (1) using RO bending angle and MWR brightness temperature measurements instead of RO refractivity and MWR TPW retrieval; and (2) overcoming the non-linearity of sequential 1DVar by using the RO and MWR observations simultaneously. The GNSS-RO bending angle obervations are simulated by the Abel integral, while the MWR brightness temperature observations are simulated by the Radiative Transfer Model (RTM) that considers atmospheric absorption, and oceanic surface reflection and emission. Our results show that this approach can simultaneously remove the GNSS-RO temperature and water vapor retrieval bias, reduce the errors in the a-priori profiles used in the retrievals, and capture the small-scale vertical structure that MWR cannot resolve. In Sec.2 the joint 1DVar retrieval algorithm using both GNSS-RO/MWR observations will be described in detail. Simulations, sensitivity tests, and actual data from collocated COSMIC-2 GNSS-RO and ATMS measurement onboard Suomi-NPP will be discussed in Sec. 3. A conclusion will be given in Sec. 4.

## 2 Joint Retrieval Algorithm: Observations, Forward model and 1D Variational method

### 2.1 GNSS-RO

In this study, the retrieved 1D bending angle $\alpha(a)$, where $a$ is the impact parameter, is used as the GNSS-RO observation. It can be related to the atmospheric refractivity through the forward Abel equation (Fjeldbo et al (1971)):

$$\alpha(a) = -2a \int_a^\infty \frac{1}{n} \frac{dn}{dx} \frac{dx}{\sqrt{x^2 - a^2}} \tag{1}$$

where $n$ is the refractive index, $x = rn(r)$, $r$ is the distance from the center of curvature to a point along the ray path, and $a$ is defined as the $x$ at the location of tangent points. The 1D bending angle is chosen as the observation in this approach for two reasons. First, it is a more "raw" data product than refractivity, reducing the likelihood of additional processing errors. Bending angles are less vertically correlated and less susceptible to errors arising from the lack of spherical symmetry than refractivity. Second, Abel inversion can only be applied to non-ducting atmospheric profiles, where $dN/dr > -157$ N-units/km. Refractivity retrievals of profiles with refractivity gradient less than this critical value will be biased below the ducting layer (Xie et al , 2006); using the bending angle retrieval instead of refractivity can avoid the negative observational bias. However, it is worth stressing that under ducting conditions, the refractivity inversion is ill-posed and infinite solutions can correspond to the same bending angle measurements. Therefore, with the bending angle alone, the thermodynamic states of the atmosphere cannot be determined, and the MWR observation is utilized in our method to anchor the solution.





The refractive index $n$ as a function of height can be calculated from the atmospheric temperature and water vapor profiles (Smith and Weintraub , 1953):

$$N = 77.6\frac{p}{T} + 3.73 \times 10^5 \frac{e}{T^2} \qquad (2)$$

where $N = (n-1) \times 10^6$ is the refractivity in $N$-units, $p$ is the pressure in mbar, and $T$ is the temperature in Kelvin. In practice, Eq. 1 can be implemented by numerical integration with changing variables. However this is a time-consuming process especially in 1DVar, where the Jacobian at multiple samples must be calculated over numerous iterations. To reduce the 1DVar Jacobian calculation complexity, Eq. 1 is implemented using the approach taken by Radio Occultation Processing Package (ROPP) (Culverwell et al. , 2015) (Burrows et al , 2014) where an exponential refractivity is assumed at each layer:

$$\frac{dln(n)}{dx} = \sim 10^{-6}\frac{dN}{dx} \qquad (3)$$

Under this assumption the bending angle integration showed in Eq. 1 can be simplified. When the refractivity decreases with height as in most situations, the bending angle for each interval is given by:

$$\Delta\alpha = 10^{-6}\sqrt{2\pi a k_i}N_i \exp\left(k_i(x_i - a)\right)\left[\text{erf}\left(\sqrt{k_i(x-a)}\right)\right]\Big|_{x_i}^{x_{i+1}} \qquad (4)$$

where erf is the error function and

$$k_i = \frac{\ln\left(N_i/N_{i+1}\right)}{x_{i+1} - x_i} \qquad (5)$$

When refractivity increases with height, we assume a constant refractivity gradient within the layer and Eq. 4 is modified to:

$$\Delta\alpha = -2\sqrt{2a}10^{-6}\frac{(N_{i+1} - N_i)}{(x_{i+1} - x_i)}\left[\sqrt{(x-a)}\right]\Big|_{x_i}^{x_{i+1}} \qquad (6)$$

Currently, ROPP processing removes the portion of the profiles below the ducting layer when its refractivity gradient reaches the critical value. Here we generalize the application of Eq. 6 to the ducting cases. When ducting occurs, $x = rn(r)$ is decreasing within the ducting layer even when its corresponding refractivity decrease sharply. In this case, $k_i$ would remain negative and the slope of $N(r)$ can be approximated as constant, which is assumed in the derivation of Eq. 6. Therefore, Eq. 6 is still applicable to calculate the bending angle profile within the ducting layer. In general, this would result in a sharp peak in the bending angle profile that can be observed in occultation profiles.

## 2.2 MWR

The MWR forward operator we use in this research is based on a radiative transfer model developed at JPL for simulation, testing, validation, and calibration of microwave radiometer measurements–from instrument design through on-orbit operation. The forward model is valid for frequencies between 6-183 GHz and can simulate both imager and sounder configurations.



Oceanic surface emission is based on Meissner and Wentz (2012). For atmosphere absorption, a number of functions can be employed. Oxygen absorption is modeled based on the work of Liebe et al (1993), Rosenkranz (1998), and Tretyakov et al (2005). Water (vapor and liquid) absorption is modeled with Liebe et al (1993). Nitrogen absorption is modeled with Rosenkranz (1998). Given inputs of instrument and environmental parameters, the forward model produces the simulated $T_b$.

150     In this article the $T_b$ measurements of all 22 channels from the ATMS instrument are simulated using the forward operator described above. However, the number of channels can be reduced by using the channels that are most sensitive to the tropospheric temperature and water vapor structure and discarding the rest. For ATMS, we can focus on channels 4 to 9 (51.76 GHz - 55.5 GHz) that are most sensitive to the tropospheric temperature, and channels 17 to 22 (165.5 GHz to 183.31 GHz) that are most sensitive to water vapor (Shao et al, 2021). By reducing the number of channels from 22 to 12, the computational

155     complexity and surface property dependence can be significantly decreased, but the thermodynamic information close to the surface will also be reduced. The trade-off between the number of channels used and the corresponding retrieval accuracy needs to be further investigated.

## 2.3   1DVar

Here we generalize the RO 1DVar inversion method described in von Engeln et al (2003) to the RO/MWR joint retrieval

160     algorithm. As in most current RO processing systems, the temperature, pressure, and the water vapor pressure at each level forms the state vector $\boldsymbol{x}$ that is being estimated:

$$\boldsymbol{x} = \begin{bmatrix} \boldsymbol{T} \\ \boldsymbol{p} \\ \boldsymbol{e} \end{bmatrix} \tag{7}$$

where $T$ is the temperature, $p$ is the pressure, and $e$ is the vapor pressure in 200 m sampling. The vertical range of the state vector spanned from 0 to 10 km altitude due to most of the vapor is distributed in the bottom 10 km of the atmosphere. Each

165     state variable in Eq. 7 is sampled every 200 m over this range to approximately match the vertical resolution of GNSS-RO. The state variables can be connected to the bending angle observation $\boldsymbol{y}_{ro}$ and brightness temperature observation $\boldsymbol{y}_{mwr}$ with the forward operators $H_{ro}$ and $H_{mwr}$ as explained in Sec.2.1 and Sec.2.2:

$$\boldsymbol{y}_{ro} = [\boldsymbol{\alpha}] = H_{ro}(\boldsymbol{x}) \tag{8}$$

$$\boldsymbol{y}_{mwr} = [\boldsymbol{T_b}] = H_{mwr}(\boldsymbol{x}) \tag{9}$$

170     where $\boldsymbol{\alpha}$ and $\boldsymbol{T_b}$ are the RO bending angle and MWR brightness temperature observations over the range of impact parameter and 22-channel frequency band respectively. As an optimal estimation approach, 1DVar will look for the best solution $\boldsymbol{x}$ by minimizing the cost function:

$$J = \frac{1}{2}(\boldsymbol{x} - \boldsymbol{x}_b)^T \boldsymbol{B}^{-1}(\boldsymbol{x} - \boldsymbol{x}_b) + \frac{1}{2}(\boldsymbol{y} - H[\boldsymbol{x}])^T \boldsymbol{O}^{-1}(\boldsymbol{y} - H[\boldsymbol{x}]) \tag{10}$$



where $\boldsymbol{y} = (\boldsymbol{y}_{ro}, \boldsymbol{y}_{mwr})$ are the RO and MWR observations, $H[x] = (H_{ro}[x], H_{mwr}[x])$ is the forward operator, and $\boldsymbol{x}_b$ are the background state variables, or a-priori, which can be obtained from a global weather analysis such as NCEP, MERRA-2, or ECMWF. $\boldsymbol{B}$ is the error covariance matrix of the a-priori $\boldsymbol{x}_b$. $\boldsymbol{O}$ is the error covariance matrix of $\boldsymbol{y}$ which includes observational and model representation errors. In this research, $\boldsymbol{O}$ is given in a diagonal form with estimated uncertainties of bending angle of $\sigma_\alpha = 8 \times 10^{-4}$ rad, and brightness temperature of $\sigma_{T_b} = 0.25$ K. The chosen bending angle uncertainty is comparable to the one of COSMIC-2 at $\sim$5km impact height (Todling et al , 2022), which represents the average RO observation uncertainty over lower troposphere. Actual RO bending uncertainty can be much smaller in free troposphere (>4km) and provide better quality observations (Todling et al , 2022). Therefore the simplified constant uncertainty setting over impact parameters shows the worst case of combination above PBL. For simplicity, the state covariance matrix $\boldsymbol{B}$ also has a diagonal form that incorporates the uncertainty of the background state variables ($T : 2.5K$, $p : 1\%$, $e : 40\%$). The background uncertainty of $T$, $p$, $e$ are chosen according to the ECMWF model used in von Engeln et al  (2003), and a similar amount of uncertainties are also assumed for other weather models. The use of a diagonal form for both covariance matrices implies the independence of measurements errors with respect to height. In reality this is not entirely true, and a more general form of covariance matrices can be found in Healy  (2001). The uncertainty defined in $\boldsymbol{B}$ and $\boldsymbol{O}$ will affect the optimal solution of temperature and vapor retrievals. More discussions on this will be provided in Sec. 3.1.

The estimated state variable $\boldsymbol{x}$ can be determined by iteratively solving the following formula:

$$\boldsymbol{x}_{n+1} = \boldsymbol{x}_b + \boldsymbol{G}_n \left[ (\boldsymbol{y} - H[\boldsymbol{x}_n]) - \boldsymbol{K}_n (\boldsymbol{x}_b - \boldsymbol{x}_n) \right] \tag{11}$$

where $\boldsymbol{x}_n$ is the state variable at the $n$-th iteration and $\boldsymbol{x}_b$ is the state a-priori. $\boldsymbol{K}_n$ is the state Jacobian matrix which can be calculated by perturbing each individual variable in $\boldsymbol{x}$:

$$\boldsymbol{K}_n = \left. \frac{\partial H(\boldsymbol{x})}{\partial \boldsymbol{x}} \right|_{\boldsymbol{x} = \boldsymbol{x}_n} \tag{12}$$

In this work we perturb T and p by 0.1K and 0.1hPa respectively, and water vapor by 0.02hPa. The gain matrix $\boldsymbol{G}_n$ can then be calculated from:

$$\boldsymbol{G}_n = \left( \boldsymbol{B}^{-1} + \boldsymbol{K}_n^T \boldsymbol{O}^{-1} \boldsymbol{K}_n \right)^{-1} \boldsymbol{K}_n^T \boldsymbol{O}^{-1} \tag{13}$$

Here we set the conditions $T_n - T_{n-1} < 0.1$K and $e_n - e_{n-1} < 0.2$hPa as the convergence criteria.

## 3 Results and Analysis of the Joint Retrieval Algorithm

### 3.1 Large Eddy Simulation

An important goal of this study is to investigate the effectiveness of the joint retrieval on the planetary boundary layer (PBL). To this end, profiles obtained from the large eddy simulation (LES) (Kurowski et al, 2022) are used to validate the 1DVar algorithm. Temperature, pressure, and water vapor from LES in 3 different physical regimes are extracted to simulate the RO bending angle and MWR $T_b$ observations and regarded as the truth:



1. Shallow to deep convective cloud transition in Amazonia region: Large-Scale Biosphere-Atmosphere (LBA) experiment (Gustavo et al, 2013).

2. Shallow cumulus cloud in west Atlantic Ocean: Rain In shallow Cumulus over the Ocean (RICO) campaign (Rauber et al, 2007).

3. Marine stratucumulus region in Northeast Pacific: Dynamics and chemistry of marine stratocumulus (DYCOMS) campaign (Stevens et al, 2003).

For each case, the RO bending angle and MWR $T_b$ observations are simulated using the forward model described in Sec. 2.1 and Sec. 2.2. Here we simulate the bending angle over 0-30km range of impact height with 50m sampling to match the RO bending resolution at lower troposphere. In this paper, no thermal noise is added to either simulated observation, although the 1DVar processing takes their uncertainty into account with the observation error covariance matrix $O$ (Sec. 2.3). The temperature a-priori is derived by smoothing the true temperature profile from LES with 1km moving window. In addition, we added a constant $-2$ K bias to the a-priori temperature at every altitude to simulate a potential bias in the background profile and test if the 1DVar algorithm can remove it. The total pressure and water vapor pressure a-priori are also defined by smoothing the true profile with 3km and 1km boxcar filters, respectively. This would remove most of the small-scaled structures in the lower troposphere that typically don't show in the background profiles. For each case the implemented 1DVar algorithm is used in three different scenarios: RO only, MWR only, and a joint retrieval using RO and MWR.

The 1DVar results between the surface and 5km are shown in Figs. 1 to 3 corresponding to the three campaigns listed above, respectively. Each figure includes the temperature profiles (a), temperature difference from truth (b), water vapor pressure (c), and water vapor difference from truth (d). In the case of LBA (Fig. 1), the true $T$ and $e$ are relatively smooth with little fine-scale vertical structures, the background profiles are close to the true profiles except for the $-2$ K added to the a-priori $T$. Fig. 1(b) shows that the temperature solution of the RO-only scenario mostly follows the a-priori profile with the $-2$ K error because the 1DVar is heavily weighted w.r.t. the $T$ a-priori compared to the water vapor a-priori. The resulting RO-only water vapor retrieval shown in Fig. 1(d) is also slightly biased ($-1$ hPa) near the surface to compensate for the negative $T$ bias and yield the same refractivity. These biases are representative of the sensitivity of the derived $T$ and $e$ to the $T$ a-priori errors in a RO 1DVar-retrieval in the lower troposphere. On the other hand, the MWR-only solutions (purple dotted-dashed) appear to be less sensitive to the a-priori $T$ error with $< 1$ K difference to the truth. The results imply that, unlike RO, the MWR is capable of independently solving for $T$ and $e$, except for the small-scale structure within 500m altitude from the surface. Combining RO bending and MWR $T_b$ in the 1DVar framework discussed above, we observe that the retrieved $T$ (red solid line) is close to truth despite the $-2$ K bias that was added to the a-priori and generate detailed water vapor retrieval in Fig. 1(c) and (d) that is more accurate than either MWR or RO alone.

In the RICO case, the water vapor retrieval from the MWR-only scenario (purple dotted-dashed) shows a large error of 2 hPa at two kilometers. This is due to the low vertical resolution of the MWR measurements which miss the fine-scale structure below 2km. As a result, the 1DVar solution for MWR-only tends to follow the shape of the given a-priori (orange dotted) which was heavily smoothed with small structure removed, and correct only the bias from the T measurements. By contrast,



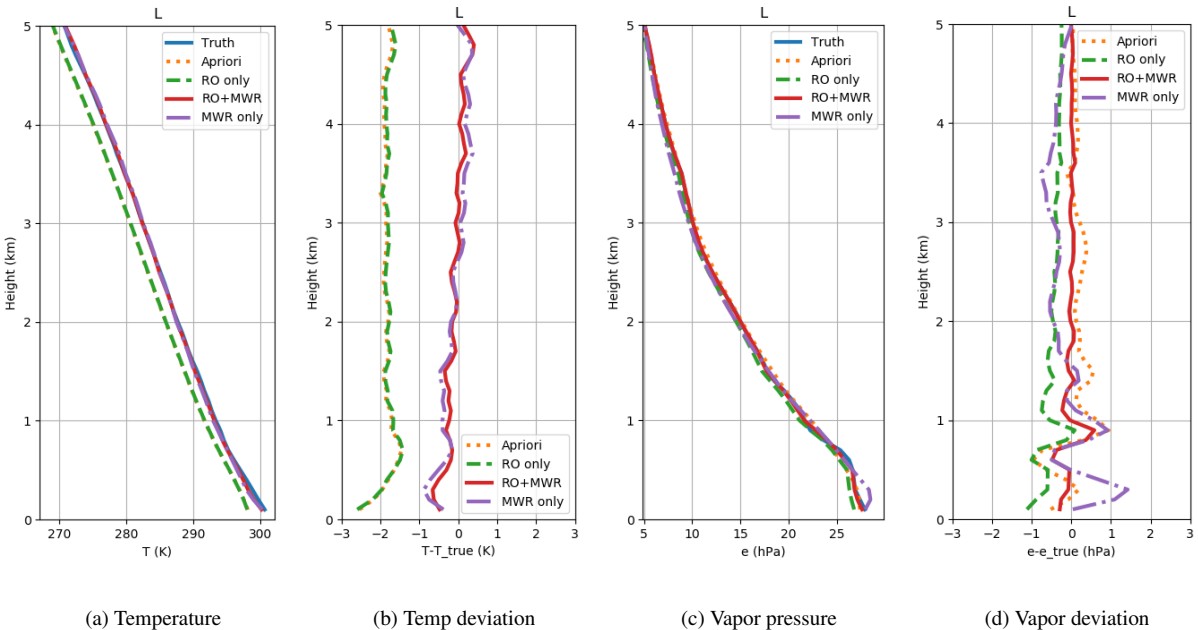

(a) Temperature      (b) Temp deviation      (c) Vapor pressure      (d) Vapor deviation

**Figure 1.** 1DVar results for LES temperature (a and b) and water vapor (c and d) profiles in LBA scheme.

the RO-only solution (green dashed) is able to reduce the water vapor retrieval error relative to the a-priori (to within 1 hPa) and better resolve the fine vertical structure. However, as shown in the LBA case the negative $e$ bias ( 0.5 hPa at the surface) persists in the lower troposphere due to $-2$ K bias in the temperature a-priori. By combining both RO and MWR with the proposed 1DVar approach, the $T$ and $e$ biases are corrected for and the small scale structures in water vapor are captured.

Fig. 3 shows the $T$ and $e$ profiles in the DYCOM case, which is located at the stratocumulus region in northeast Pacific. The lower troposphere in this region is known to have a sharp transition at the top of the boundary layer as can be seen from the temperature and moisture profiles in Fig. 3(a) and (c) at the ∼1 km height. The sharp transition of temperature and moisture create a ducting layer (Xie et al , 2006; Ao , 2007) where the RO tangent point cannot be located and the thermodynamic information in the ducting layer is lost. This results in an ill-posed inversion problem where multiple refractivity solutions would correspond to the same bending angle profile, and the standard Abel inversion would resort to a solution without ducting and cause a negative bias of up to 15% in refractivity inside the layer. This is the reason why bending angle is used in the 1DVar instead of refractivity and could, potentially avoid this refractivity bias. Nonetheless, without additional information, the solution may not converge to the correct refractivity profile in the family of solutions. As shown in Fig. 3(c) and (d), the water vapor retrieval in RO-only scenario still contain a large negative bias (-2.5 hPa close to the surface) compared to the true profile. This can be due to the -2K bias introduced in the a-priroi $T$.



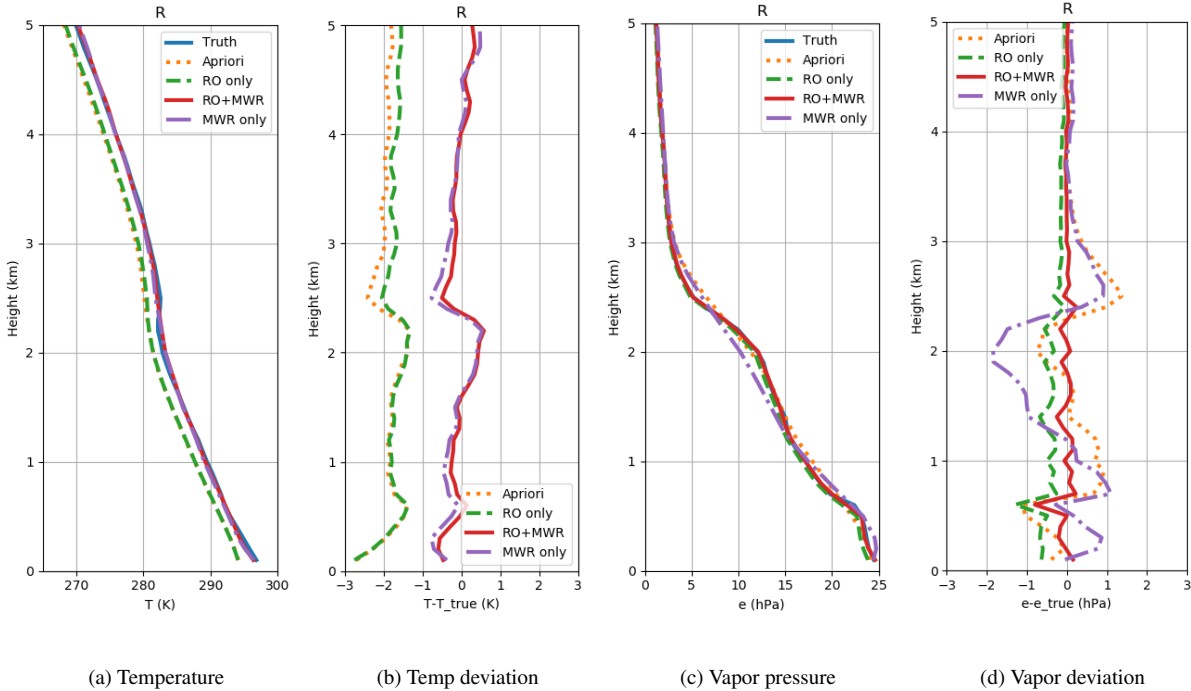

(a) Temperature      (b) Temp deviation      (c) Vapor pressure      (d) Vapor deviation

**Figure 2.** Same as Fig.1 but with RICO case

To overcome the negative $N$-bias due to ducting, Wang et al (2017) and Wang et al (2020) used the information of collocated MWR TPW retrievals and RO grazing reflected bending angles as constraints to choose the unbiased refractivity profile. Here, however, we show that combining MWR $T_b$ with the RO bending angles results in a reduced negative bias due to ducting. As shown in Fig. 3(c) and (d) the MWR only and RO/MWR scenarios are no longer limited by the ducting condition and are able to correct the moisture bias within the boundary layer. Low resolution of the MWR measurement misses the sharp transition at the PBL top, but the combination of RO and MWR preserve the advantages of both measurements. For temperature retrieval(Fig. 3(a) and (b)), While the MWR and RO/MWR solutions are able to correct the $-2$ K bias, all three solutions fail to resolve the sharp change at the transition due to strong smoothing of the background profile. With a smaller covariance of $T$ a-priori, the 1DVar temperature solution will depend more on the low resolution a-priori and MWR observations rather than the RO measurements; therefore, the large error at the transition layer should not come as a surprise.

## 3.2 Sensitivity Study

To investigate the sensitivity of the 1DVar solution to the a-priori and the measurement covariances, we perform a number of simulations using a radiosonde profile from the MAGIC campaign (magsondewnpnM1.b1.20121104.120900) (Lewis , 2016). In this study we chose a profile that does not contain any ducting conditions to avoid errors due to superrefraction. The




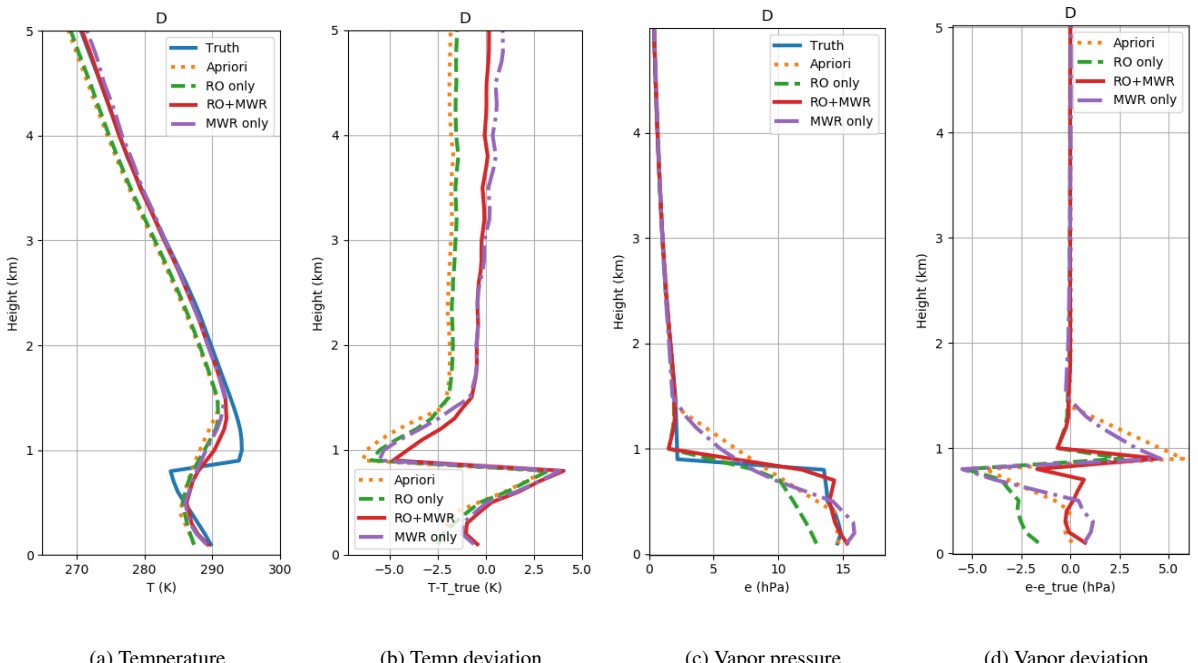

| (a) Temperature | (b) Temp deviation | (c) Vapor pressure | (d) Vapor deviation |

**Figure 3.** Same as Fig.1 but with DYCOMS case

temperature, water vapor, and the corresponding refractivity and bending angle profiles are shown in Fig. 4. The temperature and water vapor priors are obtained by a 1km boxcar filter to the RAOB $T$ and $e$ profiles. In addition to smoothing, constant biases in $T$ of $-2$ K, $-1$ K, 0 K, 1 K or 2 K were added to the a-priori $T$ profile. Alternative, constant biases in $e$ of -40%,

-20%, 0%, 20%, 40% were added to the a-priori $e$ profile. For each of these cases, three different scenarios corresponding to RO only, MWR only, and combined RO/MWR, were run. The resulting $T$ and $e$ solutions are then compared to the truth by computing the RMS error (RMSE) over the 0-5 km range. The results are shown in Fig. 5. It is clear that the RMSE for the RO-only solution (green dashed) largely follows the a-priori RMSE (orange dotted). This is expected since no independent $T$ information is provided and the 1DVar has to rely on the a-priori to estimate the $T$ profile. By contrast, the MWR-only RMSE

(purple dotted dashed) is considerably lower ($\sim$0.5K compared to the $\sim$1.8K) and is less sensitive to the $T$ bias. The combined RO/MWR scenario (red solid) reduces the $T$ RMSE even further to 0.4K and is nearly independent of the $T$ bias. On the other hand, the $e$ RMSE of the MWR-only shown in Fig. 5(b) mainly follows the a-priori due to its low resolution. While RO-only $e$ solution has lower RMSE, it is sensitive to the $T$ bias because temperature and water vapor are coupled. The combination of RO and MWR is able to uncouple the $T$ and $e$ and make the RMSE almost always below 0.3 hPa independent of the $T$

bias. By contrast, the water vapor bias has a smaller impact on retrieved temperature (Fig. 5(c)) or water vapor (Fig. 5(d)). This is mainly due to the fact that the water vapor standard deviation (set to 40%) is much larger than the temperature standard





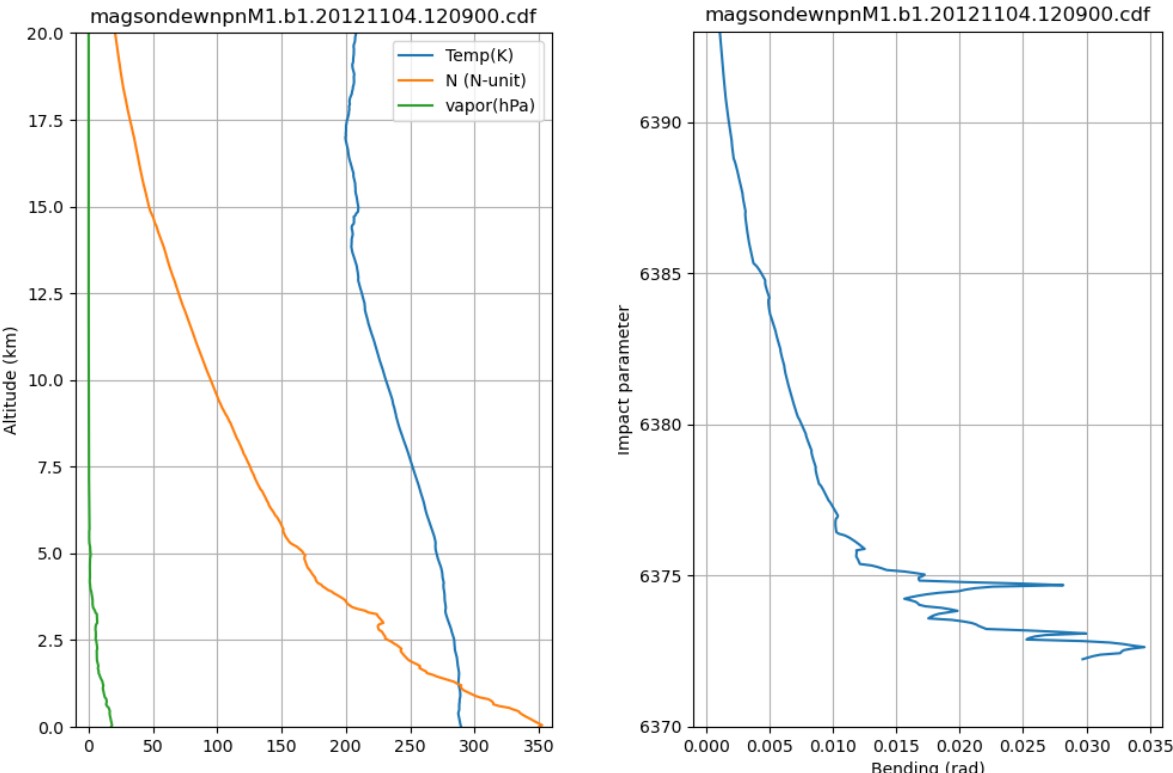

**Figure 4.** (a)Temperature, water vapor, and the refractivity profiles calculated using the Eq. 2. (b) The bending angle from the forward Abel described in Sec. 2.1

deviation (2.5K) causing the 1DVar solutions to match better with the temperature a-priori rather than the moisture a-priori. Hence the MWR-only moisture solution is not strongly biased although its structure mainly follows the water vapor a-priori. Overall, The RO/MWR combined solutions show the lowest sensitivity to the temperature and water vapor a-priori errors.

Now we turn our attention to the sensitivity of the retrievals to the MWR and RO measurement error covariances. The results are shown in Fig. 6 for 5 different levels of uncertainty in $T_b$ (0.25 K, 0.50 K, 1.00 K, 1.50 K, 2.00 K) and bending angle (0.0008 rad, 0.0012 rad, 0.0016 rad, 0.002 rad, 0.0024 rad). In all cases, the added observation have a positive impact on the solution causing the RMSE to reduce. As expected, a larger covariance reduces the value of measurements leading to larger retrieval RMSE values. This positive, near-linear trend can be seen in both MWR $T_b$ (Fig. 6(a)(b)) and RO bending angle measurements

(Fig. 6(c)(d)). In this test, no noise is added to the measurements, so that the relationship is only driven by the covariance values. Note that $T_b$ covariance should have no effects on estimates for the RO-only scenario. Similarly error covariance in bending should have no effects on estimates for the MWR-only, as confirmed in Fig. 6(c)(d). The MWR/RO combined solution has



the lowest temperature RMSE value in both scenarios (Fig. 6(a)(c)), while its moisture RMSE are almost the same as the one of RO-only solutions (Fig. 6(b)(d)). As expected, the results show that when the a-priori is not biased the MWR observation
provides little information contents to the estimation process.

The third sensitivity test shown in Fig. 7 quantifies the retrieval RMSE w.r.t a change of temperature (1.5 K, 2.5 K, 3.5 K, 4.5 K, 5.5 K) and moisture (20%, 40%, 60%, 80%, 100%) in the a-priori covariance. A higher a-priori covariance implies a greater reliance on observations in the 1DVar estimation process, and reflects different behaviors for different measurements. For the RO-only case (green curve in Fig. 7(a)) the temperature RMSE decreases under larger temperature a-priori covariance.
This may be due to a better resolved temperature profile, especially at the top of the PBL, where the strong constraint from the a-priori is relaxed. On the other hand, the MWR-only case (purple curve in Fig. 7(a)) shows the opposite trend, indicating that most of the vertical information of MWR-only retrieval is derived from the temperature a-priori. In fact, when the temperature a-priori covariance is larger than 4.5K, its corresponding RMSE exceeds the one from the a-priori itself (orange). However, the large variance observed in the temperature RMSE does not propagate to the water vapor RMSE (Fig. 7(b)) and shows a
relatively stable curve when temperature a-priori covariance changes.

It is worth noting that the perturbation of water vapor a-priori covariance does not cause dramatic change in water vapor results (Fig. 7(d)). At the same time, higher water vapor a-priori covariance would push the RO-only temperature solution closer to the a-priori profile, leading to increasing temperature RMSE as shown in Fig. 7(c). By contrast, the MWR-only temperature solutions shows negative trend which implies the independence of MWR temperature and water vapor retrieval
based on the $T_b$ measurements. As the results shown in Fig. 5 and Fig. 6, the RO/MWR joint retrieval solutions remain the lowest RMSE among all scenarios in this sensitivity test, with errors at ∼0.4K (temperature) and ∼0.3hPa (vapor pressure) in all levels. These results show that 1DVAR retrievals from combining RO and MWR combination can better reduce the error propagated from a-priori, and its solutions maintain low sensitivity to observation and a-priori covariance.

### 3.3  Real ATMS and COSMIC-2 data

We applied our joint RO/MWR 1DVar algorithm to Suomi-NPP and COSMIC-2 measurements to assess the applicability of this method to real measurements. Fig. 8 shows the collocated cases between Suomi-NPP and COSMIC-2 in a 6-hour period starting on April 1, 2019, 00:00-06:00 UTC. The Suomi-NPP data we used is the L1B calibrated/bias-corrected brightness temperature product (Lambrigtsen, 2018) provided by Goddard Earth Sciences Data and Information Services Center (GESDISC). Due to the lower inclination of COSMIC-2 orbits, all of the collocated cases are located between -45 to 45 degrees in latitude. To
illustrate the 1DVar algorithm we chose one case (2020-04-01-03:10c2f4_gps58) in the Atlantic Ocean (17.91N, 36.75W). This case is chosen because it is located on the ocean, where surface emissivity can be robustly modeled for the MWR observation. In addition, the collocated MERRA-2 reanalysis shows low ice and liquid water contents, which suggests that the MWR observation is less likely to be impacted by clouds and precipitation. Furthermore, the RO retrieval penetrated sufficiently deep in this case so that the 1DVar can estimate the thermodynamic structure within the PBL. The estimation results using different
instrument scenarios are shown in Fig. 9.





Since there is no ground truth this comparison, we cannot conclude definitely which method provides the best solution. However, by comparing the vertical structures between the different estimates we can find several clues regarding their information content.

The orange dotted lines shown in Fig. 9 (a)(b) are the profiles from the NCEP analysis, used as priors in the 1DVar processing. The green, red, and purple lines are the 1DVar solutions in the RO-only, RO/MWR, and MWR-only scenarios, respectively. The yellow line is the Community Long-Term Infrared Microwave Combined Atmospheric Product System (CLIMCAPS) retrieval (i.e., not 1DVAR) using the ATMS MWR and Cross-track infrared sounder (CrIS) data (Smith and Barnet, 2019) provided by GESDISC (Smith, 2019). Here MWR provides additional temperature information as shown in Fig. 9(a) so that the MWR-only solution (purple) deviates from the a-priori (orange) and RO-only (green) temperature retrievals by ∼3K below 2 km. The joint RO/MWR retrieval lies in between the RO-only and MWR-only solutions, and has approximately the same temperature solution as the CLIMCAPS solution. On the other hand, the water vapor profiles shown in Fig. 9(b) demonstrates that the joint RO/MWR solution is able to resolve the small-scale moisture structure throughout the profile. The deviation between the RO-only and the RO/MWR moisture solutions is caused by the temperature information that MWR provides, which in fact matches better with the ATMS moisture profile below 1km. While the improvements cannot be quantified without ground truth, the results show that the 1DVar joint retrieval combines the information and strengths of both techniques and is able to provide the high-resolution and low-bias thermodynamic profiles that a single technique cannot retrieve.

The effectiveness of combining RO and MWR observations within the framework of the proposed 1DVar approach is limited by several factors. First, the existence of cloud and precipitation can significantly increase the forward model error in $T_b$ calculation (Errico et al , 2007). Second, the MWR input and forward modeling error could still exist even in clear-sky events due to other factors that affect the surface emissivity, such as surface type, surface temperature, and surface wind speed. In this study, we limited the application of joint retrieval to an RO over the ocean. In addition, a quality control (QC) test based on the difference between the observed $T_b$ and MERRA-2 calculated $T_b$ for each channel was applied to ensure that the MWR measurements are not biased due to the reasons stated above. Statistically, 73 out of 132 colocations found within a 6-hour period passed the QC test when a RMS threshold of 10K is applied on all 22 ATMS channels.

## 4 Conclusions and Discussions

In this article, we described a 1DVar approach combining two complementary techniques to obtain high vertical resolution and solve for temperature and moisture simultaneously. Simulations were performed where LES profiles from three different campaigns were used as truth and three different scenarios corresponding to RO-only, MWR-only, and RO/MWR combination were tested. The results show that potential biases in the a-priori information used in the 1DVar can be significantly reduced after adding $T_b$ observations from MWR. At the same time, the high-resolution RO bending angle observation provides the needed vertical moisture information. The complicated thermodynamic structure in the lower troposphere, including the ones with ducting, can therefore be better resolved with much smaller biases compared to the ones using RO or MWR alone. We also analyzed the sensitivity of the temperature and vapor retrieval in each scenario to the a-priori error covariance matrix and



showed that the RO/MWR combination is the most stable among the three scenarios. Finally, the 1DVar approach is applied to real data from COSMIC-2 and Suomi-NPP observations, and the results show the promise of the 1Dvar technique.

The joint retrieval approach is similar to the optimal estimation currently used in data assimilation. Both techniques minimize the cost function, require covariance matrices to define the background and observations uncertainties, and use the forward operator to map the state vector to the observables. Typically, data assimilation for numerical weather prediction (NWP) purposes utilizes a 3D- or 4DVar approach to account for horizontal and temporal coverage. Here we only estimate the state variable on a single spatial dimension (altitude) assuming a spherically symmetric atmosphere, which reduces the number of the state variables and allows us to better quantify the relative strengths and weaknesses of the information used in the retrieval process. This also allows us to introduce as many vertical levels as needed to capture the highest vertical resolution possible with RO measurements. The proposed 1DVar joint retrieval is independent of any operational NWP model and is not limited by observation QC criteria, a-priori contribution, or the vertical grid resolution applied in operational NWP models. Therefore the joint retrieved temperature and water vapor profiles can be good candidates for validating NWP models or for providing the essential background data for potential future improvements to NWP data assimilation processes.

Two future improvements are envisioned to make the proposed 1DVar approach more practical and accurate. First, all the observation error covariance matrices we used in this study are diagonal. This implies that errors of $T_b$ of different channels and RO bending angle measurements at different heights are assumed to be independent, an assumption that is not perfectly valid. The vertical smoothing in the RO bending angle profile could lead to a high correlation of two or more neighboring samples, especially in lower altitudes. One possible way to address this issue is to use the Desrosiers diagnosis (Desroziers and Ivanov , 2001) to examine if the error covariance matrices used are reasonable, a topic that will be investigated in future studies. Second, the 1DVar approach combines RO measurements with a single set of MWR $T_b$ measurements that are nearest to the RO location. The RO location is defined as the latitude and longitude of the tangent point of the lowest link in the RO. However, in reality, RO has an extended footprint of several hundred kilometers in the occultation plan and a potential drift of the tangent point out of the occultation plane by as much as tens of kilometers. Taking horizontal variability into account could better represent the MWR $T_b$ observation at the tangent point location for each corresponding altitude. This requires a better understanding of the Jacobian function as a function of height for each channel and the location of the tangent point prior to the joint retrieval process.

*Author contributions.* K.N.W., C.O.A., and F.J.T. conceptualized the method. K.N.W. developed the methodology, and perform the analysis and validation. K.N.W. and M.G.M. implemented the software, M.J.K. provided the LES data, and C.O.A. supervised the project. K.N.W., C.O.A., G.A.H, and M.G.M., wrote the manuscript. All authors reviewed the final manuscript.





*Competing interests.* The authors declare no conflict of interest.

*Acknowledgements.* The research described in this paper was carried out at the Jet Propulsion Laboratory (JPL), California Institute of Technology, under a contract with the National Aeronautics and Space Administration. It is supported by NASA ROSES Earth Science NNH19ZDA001N-GNSS. We would also like to thank Goddard Earth Sciences Data and Information Services Center (GES DISC) for providing ATMS level-1B/2 brightness temperature and thermodynamic retrieval data product.



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





**Figure 5.** Sensitivity test of the (a, c) temperature and (b, d) vapor retrieval with respect to the temperature (a, b) and vapor pressure (c, d) a-priori







**Figure 6.** Sensitivity test of the (a, c) temperature and (b, d) vapor retrieval with respect to the $T_b$ covariance(a, b) and the bending angle covariance (c, d)





**Figure 7.** Sensitivity test of the (a, c) temperature and (b, d) vapor retrieval with respect to the temperature (a, b) and vapor pressure (c, d) covariance

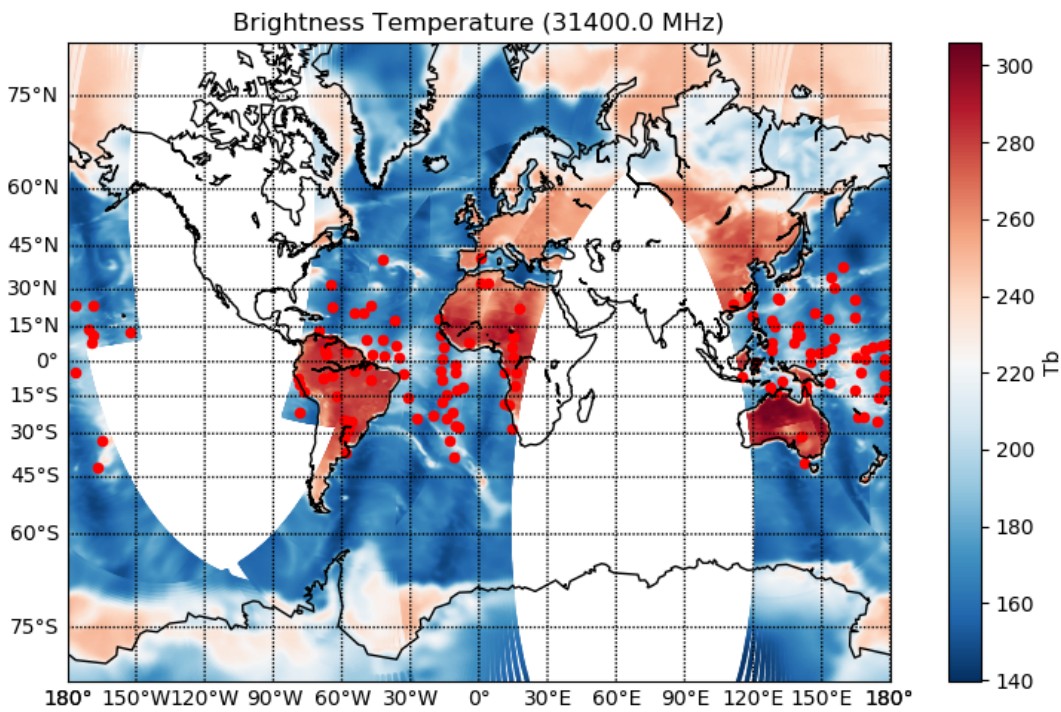

**Figure 8.** Collocated cases between Suomi-NPP and COSMIC-2 on April 1, 2019, 00:00-06:00 UTC. The red dots are the tangent point location of each COSMIC-2 RO at its lowest penetration height, while the background color is the 31.4 GHz brightness temperature measurement from the onboard ATMS instrument





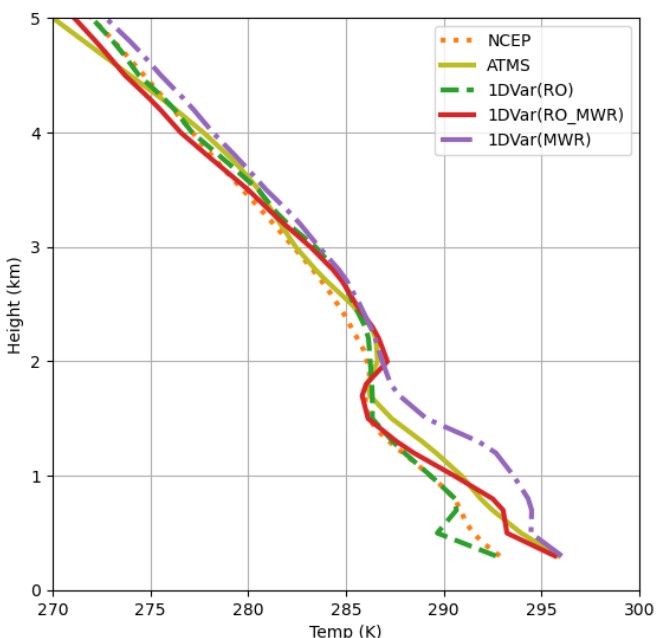

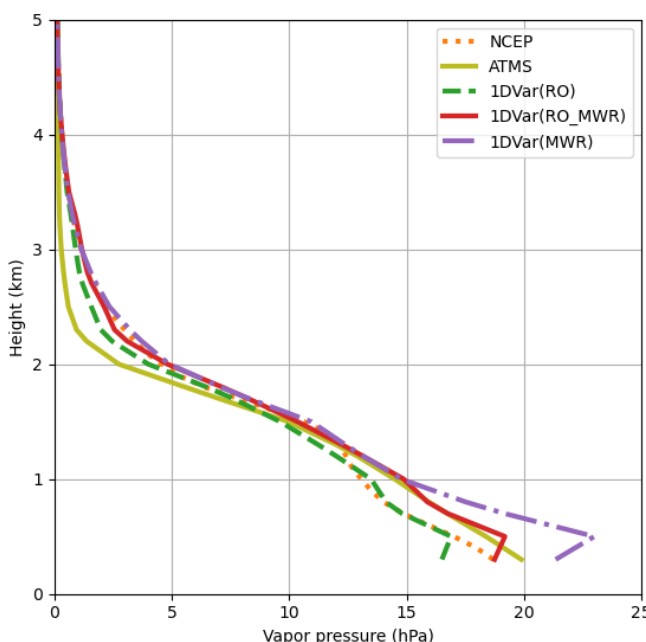

**Figure 9.** An example (2020-04-01-03:10c2f4_gps58) of the actual collocated RO-MWR combination retrieval