# Peer review of "Joint 1DVar Retrievals of Tropospheric Temperature and Water Vapor from GNSS-RO and Microwave Radiometer Observations"

_EGUsphere, 2023_

## Author Comment (AC1)

We thank the reviewer #1 and the editor for the constructive feedbacks and valuable suggestions. The followings are our responds to each reviewer #1's comment (underlined).

**General Comments:**

A nice manuscript/study, fusing two satellite observation techniques to derive an improved temperature, water vapour profile information in the lower troposphere. A few suggestions for improved readability are below.

[Authors] Thank you for the positive feedbacks, which help improve this article.

**Specific Comments:**

L10: "...characterize the complex thermodynamic structures in the lower troposphere" seems to imply this is presented in this study, but the study just proposes a way to improve the characterisation. Please re-phrase.

[Authors] We agree the vocabulary used here should be more accurate. This sentence has been modified to:

"In this study we combine these two measurement techniques in an optimal estimation approach, 1D Variation method (1DVar), to **improve the characterization of the complex thermodynamic structures** in the lower troposphere."

L53: Metop-A is no longer flying (or better, providing data), but Metop-B, -C does. So maybe just state all Metops?

[Authors] Good suggestion. We've changed it to:

"...NOAA-18, NOAA-19, **Metop series**, Advanced Microwave Scanning Radiometer..."

L68: "…they remain challenging to apply in practice" I think one major issue here is that SI traceability is lost when applying such an ad-hoc correction. Maybe that should be pointed out here too.

[Authors] We agree this is an appropriate statement to add:

"While several bias-correction methods for RO within PBL have been proposed (Xie et al , 2006; Wang et al , 2020), they remain challenging to apply in practice **and the corresponding SI traceability for refractivity could potentially be lost**."

L121: Maybe I missed it, but was the symbol e formally introduced?

[Authors] The introduction of the vapor pressure "e" is missing. It has been added back to the statement:

"p is the pressure in mbar, **e is the water vapor pressure in mbar**, and T is the temperature in Kelvin"

L164: "… state vector spanned from 0 to 10 km altitude" Suggest to add that you are ignoring the upper atmosphere in your setup, as the focus is on the lower few km and the contribution of the upper atmosphere decreases exponentially. Maybe even add an uncertainty estimate here.

[Authors] We agree that the suggested description can be added here to make the statement more convincing:

"**The vertical range of the state vector spanned from 0 to 10 km altitude. We focus on estimating the lower atmosphere because: 1) the contribution of the upper atmosphere to the lower troposphere is small due to exponentially decreases of atmospheric refractivity (with <0.4% standard deviation above 10 km), and 2) most of the vapor is distributed in the bottom 10 km of the atmosphere**."

L171: Are you using 12 or 22 channels in your MWR BT? At L154 it appears the 22 were reduced.

[Authors] In this article all 22 channels from ATMS are used. The discussion in L154 was given for possible future work plan. We make this statement clearer that 12 channels are not for current setting:

"**This process can be improved in the future** by using only the channels that are most sensitive to the tropospheric temperature and water vapor structure and discarding the rest."

"The trade-off between the number of channels used and the corresponding retrieval accuracy needs to **be further investigated in the future studies.**"

L194: Just to note that ROPP includes this differentiation, thus no need for time consuming numerical one (but of course does not have the bending angle ducting modifications included).

[Authors] Thank you for the great information! We numerically implemented the equation for all the results shown in this article, and with this ROPP precomputed Jacobian term the 1DVar will take much less time for non-ducting cases. The following sentence is added:

"**Alternatively one can use the Jacobian term calculated by ROPP software to lower the time consumption for numerical differentiation computation.**"

L229: "…to the a-priori T error with < 1 K difference to the truth" Suggest to state "… error at maximum not even 1K difference…"

[Authors] We agree this is a better statement. The sentence has been modified:

"On the other hand, the MWR-only solutions (purple dotted-dashed) appear to be less sensitive to the a-priori T **error with not even 1K difference at maximum** compared to the truth."

L234: Please add the figure you are talking about (2e?)

[Authors] In fact the figure we refer to in L234 is the Figure 1 (all panels). To better clarify this sentence has been modified:

"Combining RO bending and MWR $T_b$ in the 1DVar framework discussed above, we observe that the retrieved $T$ (red solid line) **in Fig. 1(a) and (b)** are close to truth despite the $-2$ K bias that was added to the a-priori**,** and generate detailed water vapor retrieval **(red solid line)** in Fig. 1(c) and (d) that is more accurate than either MWR **(purple dotted-dashed)** or RO **(green dotted-dashed)** alone."

L236: "…the 1DVar solution for MWR-only tends to follow the shape of the given a-priori" Suggest to add figure being discussed. And I am unsure if the "tends to follow" really captures what the MWR is showing. Seems to more show the same structure.

[Authors] Agreed.  We have addressed these comments as follows:

"In the RICO case **(Fig. 2 )**, the water vapor retrieval from the MWR-only scenario (purple dotted-dashed) shows a large error of 2 hPa at two kilometers."

"As a result, the 1DVar solution for MWR-only **shares nearly identical structure with** the given a-priori profile (orange dotted), which was heavily smoothed with small structure removed, and correct only the bias from the T measurements."

Figure 1: Left plot does not show apriori, likely covered by the green curve. And maybe add a full title, and also the figure letters a, b, c, d? As the caption talks about these, but they show nowhere on the plots. And this title point is general for all figures.

[Authors] The figures have been modified accordingly.

L265: What is this "(magsondewnpnM1.b1.20121104.120900)" exactly? A profile identifier? A file name? If name, maybe better point to where it is available.

[Authors] Thank you for the suggestion, this is the file name (.cdf) for a specific profile retrieved from the balloon-born radiosonde in MAGIC campaign. A reference is added for the data access:

**Keeler, E., & Burk, K. Balloon-Borne Sounding System (SONDEWNPN). Atmospheric Radiation Measurement (ARM) User Facility. https://doi.org/10.5439/1595321**

And the term is clarified:

"To investigate the sensitivity of the 1DVar solution to the a-priori and the measurement covariances, we perform a number of simulations using a radiosonde profile from the MAGIC campaign (**file name**: magsondewnpnM1.b1.20121104.120900)(**Keeler and Burk , 2012**)(Lewis , 2016)."

L287: "In all cases, the added observation have a positive impact..." Add weight after observation? There are no new observations added.

[Authors] To avoid confusion we changed the sentence to the following:

"In all cases, **the combination of RO and MWR observations** have a positive impact on the solution causing the RMSE to reduce."

L320: "(2020-04-01-03:10c2f4_gps58)" As above.

[Authors] This is a file name from the JPL processed COSMIC2 RO database. This case is published with the following URL:

https://genesis.jpl.nasa.gov/data/ftp/publication_data/Wang_et_al_AMT

The url of the data has also been added in the "Code and data availability" section.

Figure 6: c,d x labels not really readable.

[Authors] We changed the units of x axis from "rad" to "mrad" to make it readable.

Figure 8: The BT shown here at 31.4GHz appear not related to the channels proposed for use here (see Section 2.2: For ATMS, we can focus on channels 4 to 9 (51.76 GHz - 55.5 GHz) that are most sensitive to the tropospheric temperature, and channels 17 to 22 (165.5 GHz to 183.31 GHz) that are most sensitive to water vapor (Shao et al, 2021).) This channel shown appears to be a window one.

[Authors] We added another sub figure (Fig. 8b) using the 165.5 GHz to better reflects the channel contributes to the water vapor retrieval.

**Editorial:**

L79: "the the"

L259: ", While"

L284: "Overall, The"

[Authors] Thank you very much for catching the errors. They have been corrected.

---

## Author Comment (AC2)

We thank the reviewer #2 and the editor for the constructive feedbacks and valuable suggestions. The followings are our responds to each reviewer #2's comment (underlined).

This paper requires major revision. The authors present a 1D-Var retrieval that combines GNSS-RO bending angles and ATMS radiances with background information. A key result is that combining these two measurement techniques is better than either individually. This result is to be expected. ATMS radiances tend to provide more temperature information in the troposphere and the authors state they can "anchor the solution" (line 118). I think this is misleading because in practice the ATMS radiances will be bias corrected, to account for systematic observation and forward model errors. The calibration/bias correction of the radiances used in the 1D-Var retrieval needs to be discussed in some detail.

[Authors] Thank you for the excellent comments. In Line 118 we specifically referred to the refractivity (and subsequent temperature/water vapor) determination under ill-posed RO inversion condition when ducting occurs. Due to its closed link to the atmospheric thermodynamic states, the ATMS radiance observations do provide the additional constraint needed for selecting the correct solution from infinite number of candidate solutions.

The ATMS radiances used in this research are calibrated and bias-corrected products published by GESDISC. In this product the uncertainty caused by blackbody, cold calibration, and instrument errors have already been accurately modeled and removed from the calibration process stated in the "Algorithm Theoretical Basis Document" accompanied with the published dataset. We summarized its calibration process in Sec. 3.3:

"**This radiance dataset has been calibrated by the in-flight ATMS antenna/receiver systems that measures the radiation from two calibration sources during every scan cycle. One is the cosmic background radiation from space (cold space), and the other is the internal blackbody calibration target (hot target). By taking the radiometric counts measurements from both sources and combined with instrument errors that has been accurately modeled from ground thermal-vacuum tests, the published TB measurements are calibrated and bias-corrected. The details of in-flight calibration process are documented in the "Algorithm Theoretical Basis Document" accompanied with the published dataset. Here we assume the TB observations are unbiased and can be directly used for 1DVar combination without additional calibration steps. "**

We acknowledge that in practice the remnant error from observations, forward model, and background data could still exist and contribute to the inconsistencies between modeled thermodynamics and simulated brightness temperature. These additional errors and their impacts to the 1DVar retrieval may subject to different altitude ranges, surface properties, precipitation, and cloud particles in the atmosphere. Therefore, the results shown in Sec. 3.3

does not apply any additional calibration and/or bias correction processing after retrieving from the dataset, but "a quality control (QC) test based on the difference between the observed Tb and MERRA-2 calculated Tb for each channel was applied to ensure that the MWR measurements are not biased" (L380). The discussion of the extra uncertainty is beyond the scope of this article but has been briefly added in the following paragraph:

"**Alternatively, one can also implement an additional calibration process to further remove these factors from the MWR data and forward modeling errors. We expect this approach will improve the quality and quantity of data available for the joint retrievals, but this requires more studies in the future to statistically validate its effectiveness and uncertainty.**"

We also changed the sentence in Line 118 to clarify the idea:
"Therefore, with the bending angle alone, the thermodynamic states of the atmosphere cannot be determined, and the MWR observation is utilized in our method **to provide essential information needed for constraining the refractivity retrieval under ducting condition**.'

In addition, Collard and Healy (2003) demonstrated that RO temperature information falls as we move closer to the surface, so does this work tell us anything new about the information content of RO measurements?

[Authors] For GNSS-RO measurements, one big difference between our approach and [Collard and Healy, 2003] is that we used the rawer "bending angle" as the RO observation, while in [Collard and Healy, 2003] the retrieved "refractivity" is used. Using bending angle can avoid the biased refractivity results caused by Abel inversion when vertical refractivity gradient is high (which usually happens in the lower troposphere), and the temperature information is theoretically better preserved without additional retrieval processing. While this change can significantly reduce the number of non-converging cases due to large inconsistency between RO and MWR observations, the existing RO temperature information will not improve much when closed to the surface. This is quantified in the sensitivity study (Fig. 6) that the RO/MWR combined temperature solution RMSE has only slight increase (0.05K) even when 3x higher covariance value is applied. The discussion is also added in the paragraph describing the sensitivity test results:

"**It can be observed that the temperature RMSE (Fig. 6(a)(c)) of RO+MWR solution (red solid lines) increased by 0.22K when Tb covariance increases from 0.25 to 2 K, while it only adds 0.05K and saturated when bending angle covariance increased by 3 times. This result illustrates higher temperature information content from MWR Tb relative to RO bending angle between 0 to 5 km as expected from previous studies [e.g., Collard and Healy, 2003].**"

The simulations appear technically correct, but the 1D-Var B matrix seems unrealistic. The background errors will be vertically correlated. A 2.5 K uncertainty (line 183) may have been appropriate in 2003, but it is not now. Similarly a -2 K bias in the a priori over the entire profile  (line 215) is not realistic. A short-range forecast from NCEP, the Met Office,  … would not contain a bias of this magnitude. Therefore, assessing the value of observations on this basis does not seem reasonable.

[Authors] We agree the errors in B-matrix used here seems large in today's standard. However, one of the main messages we would like to convey in this article is that the background model uncertainty in all possible magnitudes will have limited impact when combining RO and MWR observations. Therefore, large amount of uncertainty is preferred here to test the robustness of the joint retrieval. In fact, a wide range of temperature uncertainty are tested in the sensitivity study in Sec. 3-2 (Fig. 7) of this manuscript to analyze the impact of different model variabilities toward temperature and water vapor retrievals.

In addition, by applying -2K bias we demonstrate the RO/MWR combination can still produce reasonable results even with a background error that is larger than normal.  For example, this level of error is not uncommon at the top of PBL with sharp temperature inversion.
To illustrate this point we added the following sentences to L228:

**"Here we intentionally added -2 K to test the robustness of the joint retrieval under large amount of background uncertainty. While this magnitude of error in the model is not expected to be common, it could occur at the top of the PBL where large temperature transition occurred over a short altitude interval."**

The diagonal B matrix is used for simplicity in this article, and we agree in reality the neighboring samples of the background data could be correlated. Here we added a radiosonde case (the same case used for sensitivity test) with vertically correlated B-matrix using 0.75km scale length and compare with the original vertically uncorrelated results:

[Figure]

As shown in the result the off-diagonal elements in the B matrix effectively "smooth" both temperature and water vapor solutions. The smoothing effect is expected because temperature and water vapor between neighboring levels are correlated with the added off-diagonal elements, and similar results are already shown in [Healy, 2001]. For water vapor the impact from off-diagonal terms is not obvious due to large uncertainty of vapor at this altitude range. However, adding the off-diagonal elements would cause an artificial structure in temperature above PBL for both RO and RO+MWR cases. This is mainly due to the conflict of combining smooth state variable solutions with sharp bending angle observations. This topic deserves a more careful analysis and will be addressed in a follow-on study. In the paper, we note the limitation of the current study as following:

**"So far we use the diagonal covariance matrix by assuming the model data is independent for all levels, but in reality the temperature and water vapor at neighboring levels may be correlated. To investigate the impact of the off-diagonal elements we repeat the RAOB simulation using the covariance matrix computed by the equation documented in [Healy, 2001]:**

$$B_{i,j} = \sigma_i \sigma_j exp\left[-\frac{(z_j - z_i)^2}{l^2}\right]$$

**where $i$ and $j$ are the column and row index of the background matrix respectively, $\sigma_i$ is the standard deviation at i-th level, $z$ is the level altitude, and $l$ is the scale length. In this test we set $l$ as 0.75km, and the corresponding temperature and water vapor results are shown in Fig. 8. As the figure shows the off-diagonal elements will smooth the estimated solutions with the correlation between different levels, which is consistent with the results shown in [Healy, 2001]. The enforcement of smoothness in the estimated profiles makes the combination with sharp RO bending angle observations in the lower troposphere difficult. The water vapor profile shows less small-scale structures and the RO and RO/WWR temperature profile has larger error above PBL. Therefore, while the estimation results are generally insensitive to the covariance, they could be sensitive to the correlation in the background and the off-diagonal terms have to be carefully chosen in practice. The method to select optimal off-diagonal terms in background matrix needs to be further investigated in the future."**

**"We also analyzed the sensitivity of the temperature and vapor retrieval in each scenario to the a-priori, background covariance, and observation covariance and showed that the RO/MWR combination is the most stable among the three scenarios when the background vertical levels are assumed uncorrelated."**

On line 368 it states that the retrieval is "independent of any operational NWP model", but on line 329 it says that "the NCEP analysis, used as priors in the 1DVar". This needs to be clarified.

[Authors] While the combination results show that it is much less dependent on background model, we agree the NWP model is still used here as an input and is not entirely independent. Here we rephrase the Line 368 as:

"The proposed 1DVar joint retrieval **is less dependent of any given operational NWP model** and is not limited by observation QC criteria, a-priori contribution, or the vertical grid resolution applied in operational NWP models"

In addition, did the NCEP analysis assimilate COSMIC-2 and ATMS? On line 361, the authors rightly note the similarities with data assimilation for numerical weather prediction, which is designed to retrieve information from a broad range of observation types. On 369, it states that the joint retrieval may be useful for validating NWP models. However, it is difficult to believe that is joint retrieval will be more accurate than an NWP analysis where COSMIC-2, ATMS and other observations have been assimilated. The authors will have to justify this potential NWP application more clearly before publication.

[Authors] Yes NCEP analysis does assimilate both observations. Here we meant to validate the "models" without observations, rather than NWP "analysis" that assimilated all observations. However, the combination results can assist the NWP analysis under some special circumstances. For example, the majority of COSMIC-2 cases are rejected at lower troposphere by the current QC criteria at many NWP centers, and the RO/MWR combined retrieval will then provide additional information to validate the data assimilation process results. This point is clarified to the discussion as follows:

"Therefore, the jointly retrieved temperature and water vapor profiles can be good candidates for validating **weather and climate models. In addition, under the circumstances when individual RO and/or MWR measurements are not included in the data assimilation process due to the internal QC of the NWP systems, the joint retrieval profiles can potentially provide the additional data more amenable for NWP processing.**"